# Effective interventions in preventing gestational diabetes mellitus: A systematic review and meta-analysis
Wubet Worku Takele[1], Kimberly K. Vesco[2], Jami Josefson[3], Leanne M. Redman[4], Wesley Hannah [5,6], Maxine P. Bonham[7], Mingling Chen [8], Sian C. Chivers [9], Andrea J. Fawcett [10,11], Jessica A. Grieger [12], Nahal Habibi[12], Gloria K. W. Leung[7], Kai Liu[7], Eskedar G. Mekonnen[13], Maleesa Pathirana[12], Alejandra Quinteros [12], Rachael Taylor[14], Gebresilasea G. Ukke [1], Shao J. Zhou[15], ADA/EASD PMDI* & Siew Lim [1] ✉

## Abstract

**Background** Lifestyle choices, metformin, and dietary supplements may prevent GDM, but the effect of intervention characteristics has not been identified. This review evaluated intervention characteristics to inform the implementation of GDM prevention interventions.
**Methods** Ovid, MEDLINE/PubMed, and EMBASE databases were searched. The Template for Intervention Description and Replication (TIDieR) framework was used to examine intervention characteristics (*who, what, when, where, and how*). Subgroup analysis was performed by intervention characteristics.
**Results** 116 studies involving 40,940 participants are included. Group-based physical activity interventions (RR 0.66; 95% CI 0.46, 0.95) reduce the incidence of GDM compared with individual or mixed (individual and group) delivery format (subgroup *p*-value = 0.04). Physical activity interventions delivered at healthcare facilities reduce the risk of GDM (RR 0.59; 95% CI 0.49, 0.72) compared with home-based interventions (subgroup *p*-value = 0.03). No other intervention characteristics impact the effectiveness of all other interventions.
**Conclusions** Dietary, physical activity, diet plus physical activity, metformin, and myoinositol interventions reduce the incidence of GDM compared with control interventions. Group and healthcare facility-based physical activity interventions show better effectiveness in preventing GDM than individual and community-based interventions. Other intervention characteristics (e.g. utilization of e-health) don't impact the effectiveness of lifestyle interventions, and thus, interventions may require consideration of the local context.

## Plain language summary

The effect of any given intervention to prevent gestational diabetes (high blood sugar levels that arise during pregnancy) may depend on the way it is delivered (how, when, what, etc.). This study reviewed published literature to investigate if the effects of interventions (diet, exercise, metformin, probiotics, myoinositol) to prevent gestational diabetes differ according to the way it is being delivered (e.g., online vs in-person, by health professionals or others, etc.). Exercise delivered to group settings, or those delivered at a healthcare facility worked better to prevent gestational diabetes. Although we did not observe any differences with other delivery characteristics (e.g., online vs in-person), it does not mean they are always equally effective, it is important to consider individual situations when prescribing or developing interventions.

Gestational diabetes mellitus (GDM) is a metabolic disorder characterised by hyperglycemia, usually detected by screening in the late second or early third trimester of pregnancy[1]. In 2021, the International Diabetes Federation indicated that the global prevalence of GDM was 14%[2]. GDM poses several maternal health complications, including an increased risk of pre-eclampsia, caesarean delivery, and labor induction[3]. Offspring exposed to GDM in utero are more likely to be large-for-gestational-age[4–6] and to develop impaired glucose metabolism and youth-onset type 2 diabetes[7]. Women with GDM have a risk of recurrent GDM in subsequent pregnancies[8] and have an extremely elevated lifetime risk of developing type 2 diabetes mellitus[9,10].

Although the etiology of GDM is idiopathic and multifactorial, it is presumed to be attributable to non-modifiable risk factors such as a previous history of GDM, advanced maternal age (>35 years), and family history of

A full list of affiliations appears at the end of the paper. *A list of authors and their affiliations appears at the end of the paper. ✉e-mail: siew.lim1@monash.edu

diabetes, and modifiable factors such as higher body weight[11], metabolic syndrome[12], and unhealthy lifestyle behaviors, including poor diet and lack of physical activity[13].

Maintaining a normal body mass index (BMI) during preconception and interpregnancy periods, as well as limiting excessive gestational weight gain in early pregnancy, may help reduce the risk of developing GDM in some women. For women at higher risk of GDM, interventions with lifestyle modifications (diet and physical activity), medications(metformin), and dietary supplements (probiotics and inositol/myoinositol) that promote weight loss and/or improve insulin sensitivity could play a pivotal role in minimizing its development[14–16]. Previous studies on the effects of these interventions for reducing the risk of GDM, however, have reported inconsistent findings[17–29]. Taken together, these inconsistent findings could be due, in part, to the different intervention modalities that were delivered across trials.

According to the Consolidated Framework Implementation Research (CFIR), the implementation of a program requires the identification of core components that are essential to intervention efficacy, and peripheral components that can be adapted according to the context[30]. The Template for Intervention Description and Replication (TIDieR) checklist can be used to identify the core and peripheral components across intervention characteristics, such as who conducts the intervention and where the intervention delivery occurs[31,32]. Previous systematic reviews in the general population have found that intervention characteristics such as a greater number of sessions and interventions delivered by health professionals reduce the incidence of type 2 diabetes mellitus[33] and promote weight loss in postpartum women[34]. Similarly, a meta-analysis[35] and a randomized control trial[36] demonstrated that other intervention characteristics, including lifestyle interventions assisted by technology and delivered at healthcare facilities, reduced the incidence of GDM. To date, there is no systematic review and meta-analysis that comprehensively evaluates the role of intervention type and characteristics on the effectiveness of lifestyle interventions, metformin, and dietary supplements in preventing GDM. A clear understanding of these moderating factors is essential to translate evidence from efficacy studies to implementation[32,37–42].

This review is written on behalf of the American Diabetes Association (ADA)/European Association for the Study of Diabetes (EASD) precision Medicine in Diabetes Initiative (PMDI) as part of a comprehensive evidence evaluation in support of the 2nd International Consensus Report on Precision Diabetes Medicine[43]. This study therefore aimed to investigate the effect of intervention characteristics on GDM prevention using the TIDieR framework to inform the implementation of precision prevention in healthcare and community settings.

This study identifies that dietary, physical activity, diet plus physical activity, metformin, and myoinositol interventions reduce the incidence of GDM compared with control interventions. Group and healthcare facility-based physical activity interventions show better effectiveness in preventing GDM than individual and community-based interventions.

## Methods
The Preferred Reporting Items for Systematic Review and Meta-analysis (PRISMA) 2020 guideline was used to report this study[44]. The protocol was registered in the International Prospective Register of Systematic Reviews (*PROSPERO: CRD42022320513*).

### Information sources and search strategy
Embase (Elsevier) and Ovid Medline/PubMed databases were searched to identify intervention studies published from inception through to May 24, 2022. Search strategies were built using several key terms and phrases by a professional medical librarian (AF) in consultation with the authors (SL, JJ, KV, and LR). The search was restricted to human studies and the English language. Search strategies for the respective databases are presented in Supplementary Data 1. A hand search was conducted on the reference lists of relevant reviews. All studies were exported to EndNote version 20 (Clarivate), and duplicates were identified and removed.

### Study selection procedure and eligibility criteria
The retrieved articles from several databases were exported to Endnote Version 20 (Clarivate), and duplicates were removed. Hand searches, including the reference list of related reviews, were also assessed for additional eligible studies. Covidence (Veritas Health Innovation, Melbourne, Australia), an online software, was used for title/abstract screening and full-text reviews. Randomized Controlled Trials (RCTs) and Non-Randomised Controlled Trials (Non-RCTs) were included. Editorial letters, commentary articles, and conference abstracts were excluded. Interventions included lifestyle (diet and/or physical activity), metformin, and dietary supplements (myoinositol/inositol and probiotics). Control groups were usual care/placebo or minimal intervention (no more than one lifestyle session). The primary outcome was the development of GDM. The description of eligibility criteria on the population, intervention, control, outcome, and types of study are provided in Supplementary Table 1. Two reviewers from the reviewers' team (WWT, SL, JG, MC, NH, GGU, GL, SJZ, RT, MP, KL, MB, and AQ) independently screened each record for eligibility, and disagreements were resolved by discussion with an arbiter (SL).

### Assessment of risk of bias
The quality appraisal was performed using the Cochrane Risk of Bias tool for Randomized Trials (ROB 2.0)[45] and the Risk of Bias in Non-randomized Studies of Interventions (ROBINS-I)[46] for the study type, as their name suggests. The quality of cluster RCT studies was evaluated by the ROB 2.0 tool. The ROBINS-I tool was used to assess the quality of non-RCTs. The risk of bias was assessed independently by two reviewers, and discrepancies were resolved by consensus.

### Assessment of evidence certainty
The certainty of the evidence was assessed using the Grading of Recommendations, Assessment, Development, and Evaluation method (GRADE)[47]. Five domains, namely the risk of bias (assessed using tools mentioned above), inconsistency, indirectness, imprecision, and publication bias, were used to evaluate the degree of certainty. The quality of evidence was ranked as high, moderate, low, or very low based on the GRADE guideline[48].

### Data extraction
The outcome variable (GDM incidence) was independently extracted by two reviewers. The study (authors name, study year, setting, design, and sample size) and intervention characteristics (e.g., type of intervention and intervention provider) were extracted using the TIDieR checklist[49]. The intervention characteristics include: (i) who (intervention providers/facilitators); (ii) tailoring (individualized plan); (iii) why (utilization of theoretical framework/model); (iv) how (application of technology and intervention modality); (v) what (intervention type e.g. diet, intervention material and procedure, control description); (vi) where (location of the intervention delivered; (vii) how much (duration and frequency of sessions), and (viii) how well was the intervention delivered (fidelity and attrition)[49]. Two authors (SL and WWT) independently coded the intervention characteristics, and disagreements were resolved by discussion. The detailed definition of each intervention characteristics (TIDieR constructs) is provided in Supplementary Data 2. Multiple reports from the same trial were considered as a single study.

### Data synthesis and analysis
The outcome was GDM incidence. The data were analysed using STATA/SE ™ Version 17. Risk ratios (RR) and 95% confidence intervals (CI) were pooled using the random-effects model by applying the DerSimonian and Laird estimator[50].

Heterogeneity was examined by the $I^2$ statistic[51]. Sensitivity analysis was carried out by excluding non-RCTs assuming the study design could impact the risk estimate due to lack of randomization[52]. Subgroup analysis by intervention characteristics was performed. A funnel plot

and Egger's test were used to examine publication bias. Asymmetry of the funnel plots and significant Egger's test ($p < 0.05$) suggest publication bias.

## Reporting summary

Further information on research design is available in the Nature Portfolio Reporting Summary linked to this article.

## Results

### Study selection

A total of 10,347 studies are retrieved, and 116 studies involving 40,940 participants are included. The PRISMA flow diagram is shown in Fig. 1.

### Characteristics of the included studies

A description of the included studies is shown in Supplementary Data 3. Of those included, 102 (87.9%) were RCTs. A total of 92 (79.3%) studies involved lifestyle intervention, 13 (11.1%) metformin, and 12 (10.3%) examined the role of dietary supplements (myoinositol/inositol and probiotics) in preventing GDM. The criteria used for GDM diagnosis varied across the studies. The most commonly used diagnostic criterion ($n = 37$) is the International Association of the Diabetes and Pregnancy Study Groups (IADPSG). The 1999 World Health Organization (WHO) criterion (prior to WHO adopting those of the IADPSG) was reported in nine studies, Carpenter & Coustan in seven studies, and the National Diabetes Data Group in six studies. Of these, 70 (60.3%) studies were conducted in high-income countries (predominantly Europe), and 7 were conducted in low-middle-income settings.

Seven[53–58] commenced the intervention during the preconception period, of which three were on lifestyle interventions, and the remaining were metformin interventions. The sample size ranged between 31[59] and 4,631[60] participants. The median age and BMI of participants at baseline were 30.3 years and 28.6 kg/m², respectively.

### Risk of bias and evidence quality assessment findings

Of the 102 RCTs, a high risk of bias was observed in 33 (32.4%), mainly owing to deviation from the intended intervention. Most studies (91.2%) had a low risk of bias in measuring the outcome domain. Generally, based on the overall quality judgment criterion, 33(32.4%) and 21(20.6%) of studies exhibited high and low risk of bias, respectively. In the non-RCTs, most had a low risk of bias due to the selection of study participants and reported results. A critical risk of bias due to confounding was observed in a third (33.3%) of studies. Overall, four non-RCTs were at critical risk of bias, according to the overall risk of judgment (Supplementary Data 4).

While the quality of evidence on diet-only and physical-only interventions was moderate, it was low for combined interventions (physical activity and diet). The quality of evidence for metformin, myoinositol/inositol, and probiotic interventions was very low. The most frequent reason to downgrade the level of certainty was a risk of bias and publication bias (Supplementary Data 5).

### Effect of lifestyle intervention in reducing the incidence of GDM by intervention characteristics

Supplementary Data 6 shows the characteristics of the included studies by the TIDieR framework. Of the 92 included studies investigating lifestyle intervention, 59(64.1%) included combined physical activity and dietary interventions, 17(18.5%) were physical activity-only, and 16(17.4%) were diet-only interventions. Of the studies that included a dietary intervention, nine focused on specific dietary approaches, including the Mediterranean diet and low glycaemic index diet[61–66], whilst the remaining provided general healthy dietary advice based on national dietary guidelines.

With regards to the delivery of the intervention, health professionals (e.g., dietitians, obstetricians, exercise physiologists, etc.) facilitated the intervention in 66 (71.7%). Twenty-two (23.9%) studies applied theoretical or behavioral change models, including social cognitive theory[67–74]. Except for three studies, a detailed description of the nature and procedure of the

**Fig. 1 | PRISMA flow diagram of the study.** The diagram illustrates the procedure followed to identify the eligible studies. Studies were excluded in each critical screening step based on the eligibility criteria.

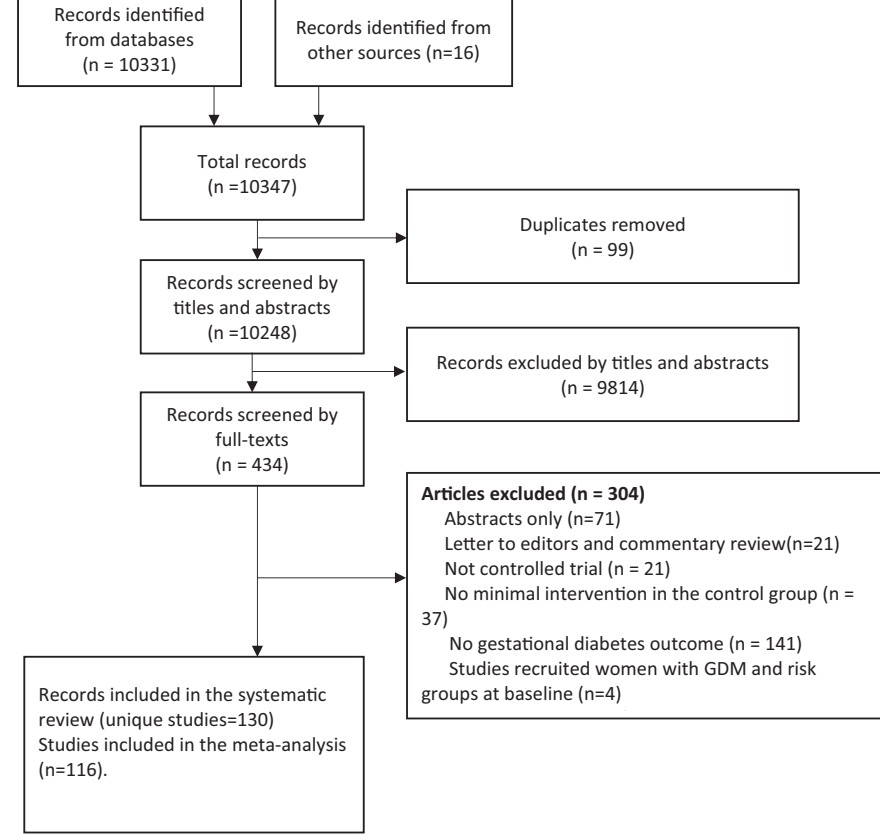

intervention delivered to the participants was reported. The care given to participants assigned to the control groups was described in 76 (82.6%) of studies. Most studies did not provide clear information on when the intervention commenced or ended for the participants[63,75–77] nor the frequency of sessions.

E-health technologies (e.g. telephone calls, WeChat, and email) were used in 46 (50%) of studies to deliver the intervention. Four studies (4.3%) provided the intervention virtually, while 49 (53.3%) delivered face-to-face only. The intervention was delivered to individuals in 17 (18.5%) studies, in group format in nine (9.8%) studies, and in combined (group and individual) in 17(18.5%) studies, while there was no description format in the rest of the studies. Three studies (3.3%) initiated the intervention during the preconception period, whereas 28 (30.4%) were in the first trimester and 58 (63%) were during the second trimester. Seventy-four (80.4%) studies utilized interventions based on individualized plans. Forty (43.5%) studies applied intervention fidelity measures, such as a curriculum for lifestyle intervention. The attrition rate of the studies ranged between 0%[78–87] and 49.3%[88].

## Table 1 | Sub-group analysis of overall lifestyle intervention by intervention characteristics

| Intervention characteristics | Number of studies | Risk ratio (95% CI) | Heterogeneity ($I^2$) | *p*-value |
|---|---|---|---|---|
| Intervention provider | | | | |
| Intervention types | | | | 0.32 |
| Physical activity | 17 | 0.69 (0.55, 0.85) | 25.9 | |
| Diet | 16 | 0.75 (0.62, 0.9) | 38.8 | |
| Combined (diet and physical activity) | 59 | 0.82 (0.73, 0.91) | 46.9 | |

## Meta-analysis of the effect of intervention characteristics on lifestyle interventions

A total of 92 studies involving 31,663 participants are included in the meta-analysis to examine the effect of lifestyle intervention on reducing GDM. Overall, lifestyle intervention reduced the incidence of GDM by 22% (RR 0.78; 95% CI 0.72, 0.85; $I^2 = 45\%$).

The difference between lifestyle intervention types was insignificant (subgroup *p*-value = 0.59) (Table 1).

### Physical activity-only intervention

Physical activity-only interventions reduced GDM by 31% (RR 0.69; 95% CI 0.55, 0.85; $I^2 = 25.9\%$; moderate quality evidence) (Fig. 2) compared with control group. According to Egger's test (*p*-value = 0.23) and funnel plot (Supplementary Fig. 1), publication bias was not observed. Group-based physical activity demonstrated the greatest reduction in risk of GDM (RR 0.66; 95% CI 0.46, 0.95; $I^2 = 28.3\%$) compared with combined (individual and group) (RR 0.79; 95% CI 0.47, 1.34; $I^2 = 0\%$) and individual (RR 1.03; 95% CI 0.72, 1.46; $I^2 = 0\%$) intervention modalities (subgroup *p*-value = 0.04). Physical activity interventions delivered in healthcare facilities reduced the risk of GDM by 41% (RR 0.59; 95% CI 0.49, 0.72; $I^2 = 33.8\%$) compared with home/community-based interventions (RR 1.05; 95% CI 0.73, 1.49; $I^2 = 58.8\%$), and combined settings (home plus and healthcare facility) (RR 1.21; 95% CI 0.67, 2.18) (subgroup *p*-value = 0.03) [Supplementary Data 7].

### Diet-only intervention

Dietary intervention reduced GDM by 27% (RR 0.73; 95% CI; 0.61, 0.86; $I^2 = 31.03\%$; moderate quality evidence) (Fig. 3). According to Egger's test (*p*-value = 0.42) and funnel plot, (Supplementary Fig. 2), publication bias was not observed. Sensitivity analysis was done by excluding two non-RCT studies, and dietary intervention reduced the risk of GDM by 25% (RR 0.75; 95% CI; 0.64, 0.88; $I^2 = 23.1\%$). None of the intervention characteristics showed an effect on the effectiveness of dietary interventions in preventing GDM (Table 2).

**Fig. 2 | Forest plot depicting the effect of physical activity on reducing the risk of GDM.** The estimates of 17 studies were pooled using the random-effects model to estimate the pooled effect of physical activity intervention on reducing the risk of GDM. The overall estimate represented in diamond shape shows the effect size (risk ratio with 95% confidence interval). The square shapes in individual study suggests the effect size estimate—the bigger the shape, the larger the effect size and the reverse is true.

| Study | Treatment Yes | Treatment No | Control Yes | Control No | Risk ratio with 95% CI | Weight (%) |
|---|---|---|---|---|---|---|
| Barakat, 2012 | 0 | 40 | 3 | 40 | 0.15 [0.01, 2.88] | 0.54 |
| Barakat, 2013 | 41 | 169 | 61 | 157 | 0.70 [0.49, 0.99] | 16.05 |
| Barakat, 2014 | 5 | 102 | 5 | 88 | 0.87 [0.26, 2.91] | 2.88 |
| Barakat, 2019 | 6 | 228 | 15 | 207 | 0.38 [0.15, 0.96] | 4.54 |
| Cordero, 2015 | 1 | 99 | 13 | 133 | 0.11 [0.01, 0.84] | 1.11 |
| da Silva, 2017 | 1 | 22 | 31 | 376 | 0.57 [0.08, 4.00] | 1.18 |
| Guelfi, 2016 | 34 | 50 | 34 | 51 | 1.01 [0.70, 1.46] | 15.30 |
| Ko, 2012 | 24 | 546 | 29 | 517 | 0.79 [0.47, 1.34] | 10.42 |
| Kong, 2014 | 1 | 17 | 1 | 18 | 1.06 [0.07, 15.64] | 0.63 |
| Oostdam, 2012 | 7 | 41 | 11 | 40 | 0.68 [0.29, 1.60] | 5.13 |
| Pelaez, 2019 | 3 | 97 | 13 | 188 | 0.46 [0.14, 1.59] | 2.78 |
| Price, 2011 | 3 | 28 | 4 | 27 | 0.75 [0.18, 3.08] | 2.17 |
| Ruiz, 2013 | 16 | 465 | 30 | 451 | 0.53 [0.29, 0.97] | 8.96 |
| Seneviratne, 2015 | 4 | 33 | 2 | 35 | 2.00 [0.39, 10.26] | 1.65 |
| Stafne, 2012 | 25 | 350 | 18 | 309 | 1.21 [0.67, 2.18] | 9.08 |
| Tomić, 2013 | 3 | 163 | 14 | 154 | 0.22 [0.06, 0.74] | 2.79 |
| Wang, 2017 | 29 | 103 | 54 | 79 | 0.54 [0.37, 0.79] | 14.79 |
| **Overall** | | | | | 0.69 [0.55, 0.85] | |

Heterogeneity: $\tau^2 = 0.04$, $I^2 = 25.93\%$, $H^2 = 1.35$
Test of $\theta_i = \theta_j$: Q(16) = 21.60, p = 0.16
Test of $\theta = 0$: z = -3.41, p = 0.00

Random-effects DerSimonian–Laird model

**Fig. 3 | Forest plot depicting the effect of dietary intervention on reducing the incidence of GDM.** The estimates of 16 studies were pooled using the random-effects model to estimate the pooled effect of dietary intervention on reducing the risk of GDM. The overall estimate represented in diamond shape shows the effect size (risk ratio with 95% confidence interval). The square shapes in individual studies suggest the effect size estimate—the bigger the shape, the larger the effect size, and the reverse is true.

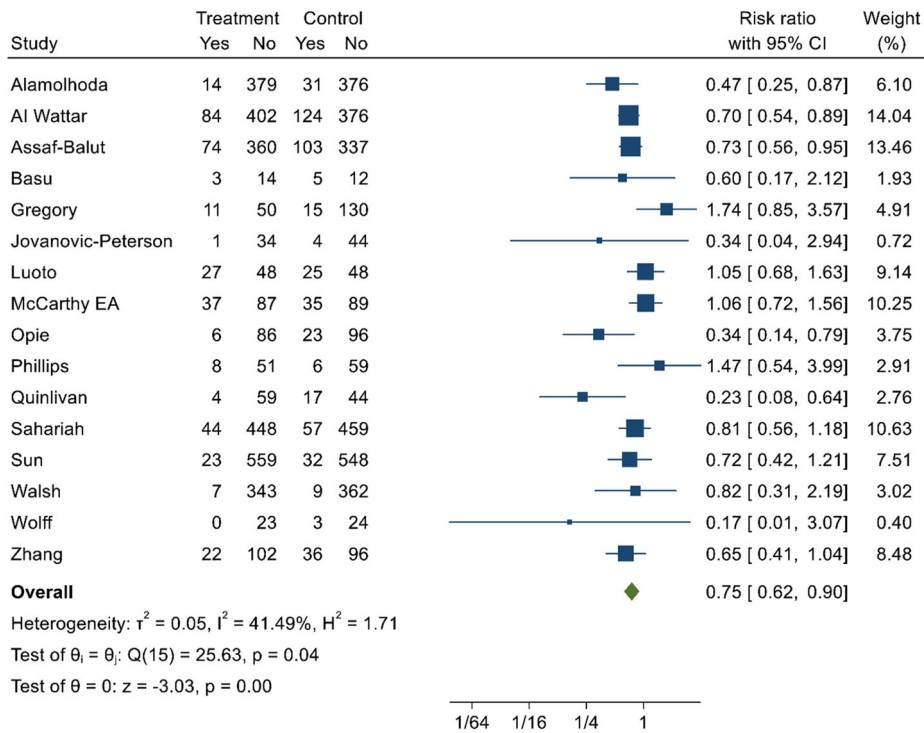

| Study | Treatment Yes | No | Control Yes | No | Risk ratio with 95% CI | Weight (%) |
|---|---|---|---|---|---|---|
| Alamolhoda | 14 | 379 | 31 | 376 | 0.47 [ 0.25, 0.87] | 6.10 |
| Al Wattar | 84 | 402 | 124 | 376 | 0.70 [ 0.54, 0.89] | 14.04 |
| Assaf-Balut | 74 | 360 | 103 | 337 | 0.73 [ 0.56, 0.95] | 13.46 |
| Basu | 3 | 14 | 5 | 12 | 0.60 [ 0.17, 2.12] | 1.93 |
| Gregory | 11 | 50 | 15 | 130 | 1.74 [ 0.85, 3.57] | 4.91 |
| Jovanovic-Peterson | 1 | 34 | 4 | 44 | 0.34 [ 0.04, 2.94] | 0.72 |
| Luoto | 27 | 48 | 25 | 48 | 1.05 [ 0.68, 1.63] | 9.14 |
| McCarthy EA | 37 | 87 | 35 | 89 | 1.06 [ 0.72, 1.56] | 10.25 |
| Opie | 6 | 86 | 23 | 96 | 0.34 [ 0.14, 0.79] | 3.75 |
| Phillips | 8 | 51 | 6 | 59 | 1.47 [ 0.54, 3.99] | 2.91 |
| Quinlivan | 4 | 59 | 17 | 44 | 0.23 [ 0.08, 0.64] | 2.76 |
| Sahariah | 44 | 448 | 57 | 459 | 0.81 [ 0.56, 1.18] | 10.63 |
| Sun | 23 | 559 | 32 | 548 | 0.72 [ 0.42, 1.21] | 7.51 |
| Walsh | 7 | 343 | 9 | 362 | 0.82 [ 0.31, 2.19] | 3.02 |
| Wolff | 0 | 23 | 3 | 24 | 0.17 [ 0.01, 3.07] | 0.40 |
| Zhang | 22 | 102 | 36 | 96 | 0.65 [ 0.41, 1.04] | 8.48 |
| **Overall** | | | | | 0.75 [ 0.62, 0.90] | |

Heterogeneity: $\tau^2 = 0.05$, $I^2 = 41.49\%$, $H^2 = 1.71$

Test of $\theta_i = \theta_j$: Q(15) = 25.63, p = 0.04

Test of $\theta = 0$: z = -3.03, p = 0.00

1/64  1/16  1/4  1

Random-effects DerSimonian–Laird model

## Table 2 | Sub-group analysis of dietary intervention by intervention characteristics

| Intervention characteristics | Number of studies | Risk ratio (95%CI) | Heterogeneity ($I^2$) | p-value |
|---|---|---|---|---|
| *Individually tailored* | | | | 0.96 |
| Yes | 9 | 0.75 (0.55, 1.03) | 55.2 | |
| No | 7 | 0.76 (0.61, 0.94) | 22.8 | |
| *Intervention modality* | | | | 0.55 |
| Individual-based | 4 | 0.96 (0.59, 1.54) | 65.1 | |
| Group-based | 1 | 0.82 (0.31, 2.2) | – | |
| Combined (individual and group-based) | 2 | 0.71 (0.59, 0.85) | 0 | |
| Unspecified | 9 | 0.65 (0.49, 0.85) | 21.8 | |
| *Application of technology (e.g. Wechat and Facebook)* | | | | 0.24 |
| Yes | 3 | 0.6 (0.41, 0.88) | 22 | |
| No | 12 | 0.79 (0.64, 0.98) | 43.2 | |
| *Fidelity* | | | | 0.2 |
| High/medium | 2 | 0.52 (0.28, 0.96) | 42.3 | |
| Low | 14 | 0.79 (0.65, 0.96) | 39.9 | |
| *Medium of delivery* | | | | 0.24 |
| In-person only | 13 | 0.79 (0.64, 0.98) | 43.2 | |
| Hybrid | 3 | 0.6 (0.41, 0.89) | 22 | |
| *Country's income level* | | | | 0.72 |
| High-income | 12 | 0.79 (0.61, 1.02) | 51.1 | |
| Upper-middle income | 2 | 0.68 (0.48, 0.96) | 0 | |
| Low-middle income | 2 | 0.65 (0.39, 1.1) | 55.2 | |

### Combined (diet and physical activity) intervention

The combined diet and physical activity intervention lowered the incidence of GDM by 18% (RR 0.82; 95% CI 0.74, 0.94; $I^2 = 46\%$; low-quality evidence). According to Egger's test (*p*-value = 0.01) and funnel plot (Supplementary Fig. 3), publication bias was observed. After excluding six non-RCTs, combined lifestyle intervention reduced the risk of GDM by 17% (RR 0.83; 95% CI 0.74, 0.93; $I^2 = 64.8\%$). Combined lifestyle interventions conducted in low-middle income countries (RR 0.51; 95% CI 0.32, 0.8; $I^2 = 17.3\%$) demonstrated a larger effect in reducing the risk for GDM than middle-income countries (RR 0.69; 95% CI 0.56, 0.83; $I^2 = 52.5\%$) and high-income countries (RR 0.93; 95% CI 0.84, 1.04; $I^2 = 52.5\%$) (subgroup p-value = 0.00) (Table 3). The incidence of GDM did not differ by any other intervention characteristic.

**Table 3 | Sub-group analysis of combined lifestyle intervention by intervention characteristics**

| Intervention characteristics | Number of studies | Risk ratio (95%CI) | Heterogeneity (I²) | p-value |
|---|---|---|---|---|
| *Intervention provider* | | | | |
| Health professionals | 46 | 0.85 (0.75, 0.96) | 46 | 0.08 |
| Non-health professionals | 10 | 0.69 (0.56, 0.84) | 0 | |
| *Individually tailored* | | | | 0.21 |
| Yes | 49 | 0.84 (0.75, 0.94) | 46.2 | |
| No | 10 | 0.69 (0.54, 0.91) | 33.6 | |
| *Intervention modality* | | | | 0.32 |
| Individual-based | 10 | 0.89 (0.69, 1.14) | 33.6 | |
| Combined (individual and group-based) | 14 | 1.01 (0.85, 1.19) | 37.1 | |
| Unspecified | 35 | 0.73 (0.63, 0.83) | 36.2 | |
| *Application of framework/theory* | | | | 0.54 |
| Yes | 21 | 0.85 (0.73, 0.98) | 26.6 | |
| No | 38 | 0.79 (0.68, 0.92) | 54.1 | |
| *Application of technology (e.g. Wechat and Facebook)* | | | | 0.24 |
| Yes | 40 | 0.86 (0.77, 0.96) | 23.1 | |
| No | 19 | 0.74 (0.59, 0.93) | 68.8 | |
| *Fidelity* | | | | 0.83 |
| High/medium | 23 | 0.81 (0.69, 0.94) | 47.5 | |
| Low | 36 | 0.83 (0.73, 0.96) | 47 | |
| *Medium of delivery* | | | | 0.17 |
| In-person only | 20 | 0.71 (0.57, 0.9) | 69.8 | |
| Hybrid | 35 | 0.89 (0.79, 0.99) | 21.9 | |
| Virtual-only | 4 | 0.73 (0.52, 1.03) | 0 | |
| *Country of the studies* | | | | 0.00 |
| High-income | 43 | 0.93 (0.84, 1.04) | 52.5 | |
| Upper middle-income | 11 | 0.69 (0.56, 0.83) | 52.5 | |
| Low-middle income | 5 | 0.51 (0.32, 0.8) | 17.3 | |

### Effect of metformin on reducing the incidence of GDM by intervention characteristics

Thirteen studies were included. Nine studies described the intervention given to participants assigned to the placebo groups, two applied tailored interventions[27,89,90], and one was technology-based (telephone)[89]. The range of daily dosage was 500[89]–3000 mg[91]. Eight studies monitored the adherence of participants to the medication through pill count. The attrition rate ranged from 0% to 42%[92]. The detailed intervention characteristics are presented in Supplementary Data 8.

On meta-analysis, metformin reduced the risk of developing GDM by 34% (RR 0.66; 95% CI 0.47, 0.93; $I^2 = 73.08\%$; very low-quality evidence) (Fig. 4). According to Egger's test ($p = 0.00$) and funnel plot (Supplementary Fig. 4), publication bias was detected. Further subgroup analysis was not undertaken due to insufficient studies on each intervention characteristics group.

### Effect of dietary Supplements on reducing the incidence of GDM by intervention characteristics

**Probiotic supplementation.** Five studies examined the relationship between probiotic supplements and the incidence of GDM. Three combined investigations of supplementation with a probiotic and another intervention (one co-administered a fish oil supplement)[93], one applied an additional unspecified dietary intervention[66], and one applied a technology via telephone[94]. Three (60%) studies monitored participants' adherence to the intervention mainly through pill counts[93–95]. The attrition rate was 2.7%[96]–25.4%[66]. A detailed description is provided in Supplementary Data 9.

On meta-analysis, probiotics supplements did not reduce the risk of GDM (RR 0.88; 95% CI; 0.52, 1.47; $I^2 = 73.7\%$; very low-quality evidence). The Eggers test ($p$-value = 0.24) and funnel plot (Supplementary Fig. 5) reveal the absence of publication bias. By intervention type, probiotics co-administered with diet (RR 0.36; 95% CI; 0.18, 0.72), probiotics alone (RR 1.0; 95% CI 0.56, 1.81), and probiotics coupled with fish oil (RR 1.3; 95% CI 0.78, 2.15) reduced the risk of GDM (Fig. 5). Subgroup analysis by the intervention characteristics was not performed due to the limited number of studies in each subgroup.

**Myoinositol/inositol supplement.** Seven studies[96–102] examined the effect of myoinositol/inositol supplements in preventing GDM (Supplementary Data 4). On the meta-analysis, myoinositol/inositol supplement reduced the risk of GDM by 61% (RR 0.39; 95% CI 0.23, 0.66; $I^2 = 78.87\%$; very low-quality evidence) (Fig. 6). Egger's test ($p$-value = 0.26) and funnel plot (Supplementary Fig. 6) exhibited that publication bias was not a concern.

Subgroup analysis by the intervention characteristics was not performed due to the limited number of studies in each subgroup.

### Discussion

In this comprehensive systematic review and meta-analysis, interventions utilizing diet, physical activity, diet plus physical activity, metformin, and myoinositol reduced the incidence of GDM compared with control interventions. The findings are in line with the most recent findings from umbrella reviews[103,104], implying the importance of incorporating these interventions in routine maternal care to prevent GDM. However, the

**Fig. 4 | Forest plot depicting the effect of metformin on preventing GDM.** The estimates of 13 studies were pooled using the random-effect model to estimate the pooled effect of metformin intervention on reducing the risk of GDM. The overall estimate represented in diamond shape shows the effect size (risk ratio with 95% confidence interval). The square shapes in individual studies suggest the effect size estimate—the bigger the shape, the larger the effect size, and the reverse is true.

Fig. 4 data:

| Study | Treatment Yes | Treatment No | Control Yes | Control No | Risk ratio with 95% CI | Weight (%) |
|---|---|---|---|---|---|---|
| Chiswick, 2008 | 36 | 117 | 26 | 116 | 1.29 [ 0.82, 2.01] | 10.56 |
| Dodd, 2018 | 72 | 184 | 62 | 196 | 1.17 [ 0.87, 1.57] | 11.83 |
| Glueck, 2002 | 1 | 32 | 10 | 27 | 0.11 [ 0.02, 0.83] | 2.36 |
| Jamal, 2012 | 3 | 32 | 6 | 29 | 0.50 [ 0.14, 1.84] | 4.45 |
| Lovvik, 2019 | 60 | 178 | 57 | 183 | 1.06 [ 0.77, 1.45] | 11.66 |
| Sales, 2018 | 13 | 69 | 16 | 66 | 0.81 [ 0.42, 1.58] | 8.66 |
| Syngelaki, 2016 | 25 | 177 | 22 | 173 | 1.10 [ 0.64, 1.88] | 9.78 |
| Valdés, 2018 | 16 | 47 | 18 | 30 | 0.68 [ 0.39, 1.18] | 9.60 |
| VANKY, 2010 | 22 | 103 | 21 | 103 | 1.04 [ 0.60, 1.79] | 9.73 |
| Adb El Hameed, 2011 | 1 | 30 | 6 | 20 | 0.14 [ 0.02, 1.09] | 2.26 |
| Ainuddin, 2015 | 5 | 45 | 11 | 21 | 0.29 [ 0.11, 0.76] | 6.37 |
| Khattab, 2011 | 8 | 192 | 32 | 128 | 0.20 [ 0.09, 0.42] | 7.97 |
| Glueck, 2022 | 3 | 65 | 9 | 25 | 0.17 [ 0.05, 0.58] | 4.75 |
| **Overall** | | | | | 0.66 [ 0.47, 0.93] | |

Heterogeneity: $\tau^2 = 0.23$, $I^2 = 73.08\%$, $H^2 = 3.71$
Test of $\theta_i = \theta_j$: Q(12) = 44.57, p = 0.00
Test of $\theta = 0$: z = -2.36, p = 0.02

Random-effects DerSimonian–Laird model

**Fig. 5 | Forest plot depicting the effect of probiotics supplements on preventing GDM.** The estimates of six studies were pooled using the random-effect model to estimate the effects of different categories of probiotics supplementation on reducing the risk of GDM. The red diamond shape shows the effect size (risk ratio) in each subgroup. The overall estimate represented in the green diamond at the bottom shows the overall effect size (risk ratio). The square shapes in individual studies suggest the effect size estimate —the bigger the shape, the larger the effect size, and the reverse is true.

Fig. 5 data:

| Study | Treatment Yes | Treatment No | Control Yes | Control No | Risk ratio with 95% CI | Weight (%) |
|---|---|---|---|---|---|---|
| **Diet+Probiotics** | | | | | | |
| Luoto, 2010 | 9 | 64 | 25 | 48 | 0.36 [ 0.18, 0.72] | 18.55 |
| Heterogeneity: $\tau^2 = 0.00$, $I^2 = .\%$, $H^2 = .$ | | | | | 0.36 [ 0.18, 0.72] | |
| Test of $\theta_i = \theta_j$: Q(0) = 0.00, p = . | | | | | | |
| **Fish oil+Probiotics** | | | | | | |
| Pellonper¨a, 2019 | 26 | 65 | 20 | 71 | 1.30 [ 0.78, 2.15] | 21.88 |
| Heterogeneity: $\tau^2 = 0.00$, $I^2 = .\%$, $H^2 = .$ | | | | | 1.30 [ 0.78, 2.15] | |
| Test of $\theta_i = \theta_j$: Q(0) = 0.00, p = . | | | | | | |
| **Probiotics** | | | | | | |
| Lindsay, 2014 | 10 | 52 | 11 | 63 | 1.09 [ 0.49, 2.38] | 16.86 |
| Wickens, 2017 | 15 | 169 | 26 | 163 | 0.59 [ 0.32, 1.08] | 20.12 |
| Callaway, 2019 | 38 | 169 | 25 | 179 | 1.50 [ 0.94, 2.39] | 22.58 |
| Heterogeneity: $\tau^2 = 0.18$, $I^2 = 64.91\%$, $H^2 = 2.85$ | | | | | 1.00 [ 0.56, 1.81] | |
| Test of $\theta_i = \theta_j$: Q(2) = 5.70, p = 0.06 | | | | | | |
| **Overall** | | | | | 0.88 [ 0.52, 1.47] | |

Heterogeneity: $\tau^2 = 0.25$, $I^2 = 73.71\%$, $H^2 = 3.80$
Test of $\theta_i = \theta_j$: Q(4) = 15.22, p = 0.00
Test of group differences: $Q_b(2) = 8.94$, p = 0.01

Random-effects DerSimonian–Laird model

primary analysis of this review has previously shown that not all interventions work equally for all participants[105], and therefore, considering person-level characteristics (e.g., previous history of GDM) during implementation could be important to enhance the effectiveness of interventions. This secondary analysis shows the differences in the intervention effectiveness by intervention type and delivery. For physical activity interventions, those delivered in groups or in healthcare facilities resulted in a greater reduction in the risk of developing GDM compared with individual and combined (group and individual) formats and with community-home-based interventions. Diet-only interventions were similarly effective across all delivery contexts. Combined diet and physical activity interventions conducted in low-middle-income countries demonstrated a greater reduction in GDM than in upper-middle and high-income countries. Insufficient data were available for meta-analysis for metformin and dietary supplements.

This analysis found that group-based delivery was a more effective delivery format for physical activity interventions compared with individual-based or individual-plus group formats. This finding is in line with a systematic review demonstrating that group-based physical activity helps prevent GDM[106]. The greater effectiveness of group-based delivery for physical activity intervention may be due to a high number of studies (76.5%) within this category providing fully supervised sessions. The finding of this study is consistent with previous systematic reviews and meta-

**Fig. 6 | A forest plot showing the effect of myoinositol/inositol on reducing the risk of GDM.** The estimates of seven studies were pooled using the random-effects model to estimate the effects of different categories of myoinositol supplementation on reducing the risk of GDM. The overall estimate represented in the green diamond at the bottom of the figure shows the overall effect size (risk ratio with 95% confidence interval). The square shapes in individual studies suggest the effect size estimate—the bigger the shape, the larger the effect size, and the reverse is true.

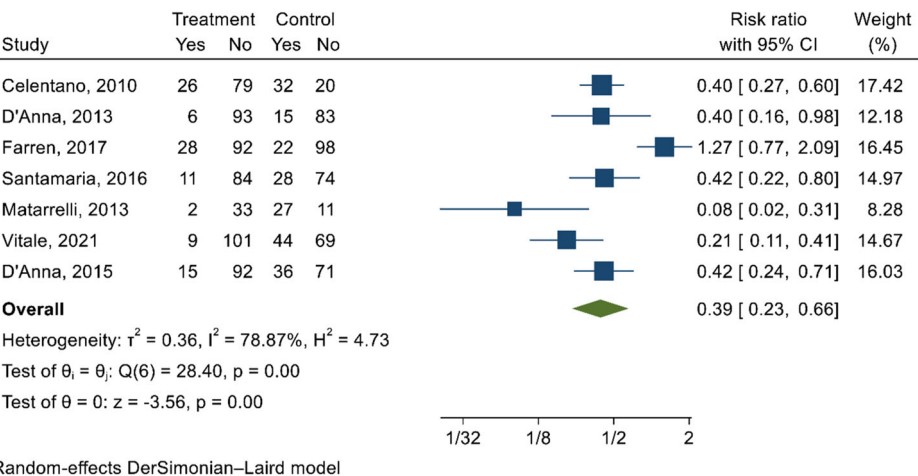

analyses in individuals with type 2 diabetes, where it has been reported that supervised physical activity intervention enhanced the effectiveness of blood glucose management in these individuals[107,108]. This may be because those studies utilized multiple behavior change techniques concurrently, including behavioral practice/rehearsal, demonstration of the behaviors, and feedback on behavior[107]. Group-based interventions may also be more beneficial in the prevention of GDM as they create the opportunity for women to engage with their peers. This occurs when individuals within a group share ideas and experiences, which could help enhance their commitment and motivation, ultimately motivating them to stay in the intervention program for the desired intervention period[109]. Greater effectiveness with group-based interventions has also been shown previously in diabetes and weight management interventions[107,110–112]. Peer support has been shown to predict physical activity behavior change in adolescents, serving as behavior change agents who provide support and role modelling to actively engage and maintain lifestyle interventions[113].

In addition, group-based intervention is an acceptable delivery format by healthcare providers and by women. A systematic review among healthcare providers reported perceived positive experiences from group-based antenatal care, including richer use of their time and better value proposition in terms of provider investment and workload[114]. A recent systematic review of qualitative studies has also found that group-based physical activity was highly acceptable by women[115]. Given the observed effectiveness of group-based physical activity during pregnancy in reducing the risk of GDM and considerable acceptability by healthcare professionals and service users, it may be beneficial to utilize this format in a real-world setting.

Delivery settings could also affect the effectiveness of physical activity interventions in preventing GDM. Those initiated during pregnancy in health facilities reduced the incidence of GDM more than home/community-based interventions. This finding is supported by previous meta-analyses of RCTs of supervised physical activity interventions during pregnancy in preventing GDM[35,116]. In-facility interventions may provide opportunities for supervision and feedback from professionals, which likely enhance the adherence of participants and as a result, improve the intervention effectiveness[117]. However, as data on the level of adherence to physical activity interventions delivered in different settings were not reported, it is impossible to draw an inference that the better effectiveness of healthcare facility-based interventions are related to the better adherence of participants to the intervention. Future primary studies are recommended to examine the role of adherence in the effectiveness of physical activity interventions delivered in different settings.

Since all the physical activity intervention studies included in our review commenced during pregnancy, our findings may not be applicable to interventions started during the preconception or postpartum period, during which additional barriers to accessing interventions may exist. A systematic review of RCTs underscored that supervised physical activity intervention during the postpartum period leads to a high rate of refusal and withdrawal from the intervention[118]. This suggests the reproductive life stages of the participants are an important consideration in the choice of intervention setting. To foster better adherence of individuals to interventions throughout the inter-conception period, healthcare facilities need to be accessible to women and provide the necessary resources such as childcare[119]. Home-based interventions could be an alternative and preferred modality for reproductive-age women due to fewer barriers, such as parenting responsibilities and time constraints[120,121]. These factors must be considered when selecting the intervention setting. A flexible approach that considers home/community-based sessions supported by virtual or in-person supervision may provide equivalent benefits to healthcare facility-based interventions. Future trials are recommended to compare the role of different intervention settings across the reproductive life stages in preventing GDM and with an evaluation of adherence rate, consumer satisfaction, and resources required to generate user-informed and sustainable evidence-based practice in real-world settings.

Moreover, differences in the effectiveness of physical activity intervention across other intervention characteristics, including intensity and type of physical activity, were not observed. Similar to a recent umbrella review[103], we found that physical activity interventions of light-moderate or moderate intensity effectively reduced the risk of GDM. However, the differences between subgroups by intensity were found to be insignificant ($p$-value = 0.18). It was evidenced that light to moderate or moderate intensity reduced the incidence of GDM compared with moderate to vigorous intensity[103]. Given the effectiveness of light-moderate activities, which are more achievable than higher-intensity training, especially during pregnancy, women at risk of GDM should be recommended to engage in moderate-intensity activities to reduce their GDM risk.

We observed that studies on combined lifestyle interventions conducted in low-middle-income countries demonstrated greater effectiveness in reducing the risk of GDM than in high- and upper-middle-income countries. Given the consistent evidence showing the effectiveness of lifestyle intervention in preventing GDM in low-middle income countries[122], along with the growing diabetes burden in this region[2,123], there is an urgent need for large-scale implementation of combined lifestyle intervention to curb the growing incidence of GDM in low-middle income countries. On the other hand, there is a paucity of studies in low-income countries, as evidenced by our study and a previous review[122], which is an evidence gap hindering the reduction of global diabetes disparities in these regions. Thus, future studies are needed in low-middle-income countries to demonstrate the effectiveness of lifestyle interventions in GDM prevention and to identify effective intervention characteristics in these.

Diet-only interventions reduced the risk of GDM irrespective of the intervention characteristics (e.g., e-health and home-based) and setting (i.e. country). This suggests that dietary interventions could be delivered in any format according to contextual needs without compromising effectiveness

in GDM prevention. However, comparison by intervention duration, frequency and dietary types were not performed due to poor reporting in the included studies, as reflected in a previous review[124].

Future individual studies should improve the reporting on these characteristics to enable further elucidation of optimal duration, frequency, and dietary type of interventions in preventing GDM.

This is the first comprehensive review that investigated intervention characteristics of lifestyle, metformin, and dietary supplements in preventing GDM. The approach is underpinned by established frameworks such as CFIR for intervention implementation[30] and TIDieR for the identification of intervention characteristics[49]. However, missingness in certain intervention characteristics (frequency of sessions and duration) was a major barrier in examining the effectiveness of these intervention characteristics. In addition, when interpreting and translating the evidence, it is important to note that substantial heterogeneity remained within subgroups, suggesting other sources of heterogeneity were present such as bias due to inclusion of non-RCTs[52]. However, the sensitivity analysis excluding non-RCTs did not alter the effect of interventions on reducing the incidence of GDM. Given the poor adherence of authors of individual studies to the evidence reporting checklist (TIDieR framework), coding was subject to interpretation. This was attempted to mitigate by having two trained reviewers (WWT and SL). Lastly, the certainty of quality of evidence for all interventions ranged from low to moderate, suggesting caution when applying the findings in real-world settings.

## Conclusions
Dietary, physical activity, diet plus physical activity, metformin, and myoinositol interventions during pregnancy reduce the incidence of GDM compared with control interventions. Group and healthcare facility-based physical activity interventions during pregnancy reduce the risk of GDM compared with individual-based and home/community-based interventions, respectively. Dietary interventions could be implemented in any format with considerations of contextual factors. Researchers conducting intervention trials better follow TIDieR guidelines when reporting to enable the identification of key components for the implementation of interventions to prevent GDM.

## Data availability
All data used to produce this study was gathered from published studies. The key terms and search strategies built to retrieve studies are available in Supplementary Table 1 of the Supplementary Information file. The list of included studies is available in Supplementary Data 1. All other relevant data that support the findings of the study are available from the corresponding author upon reasonable request.

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

## Acknowledgements

The ADA/EASD Precision Diabetes Medicine Initiative, within which this work was conducted, has received the following support: The Covidence license was funded by Lund University (Sweden), for which technical support was provided by Maria Björklund and Krister Aronsson (Faculty of Medicine Library, Lund University, Sweden). Administrative support was provided by Lund University (Malmö, Sweden), the University of Chicago (IL, USA), and the American Diabetes Association (Washington D.C., USA). The Novo Nordisk Foundation (Hellerup, Denmark) provided grant support for in-person writing group meetings (PI: L Phillipson, University of Chicago, IL) (NNF22SA0081670).SL is funded by the Australian National Health and Medical Research Council (NHMRC) Fellowship (APP1139481). JW is funded by the NHMRC Ideas Grant. WT, MC, and GU were funded by the Australian Government Research Training Program Scholarship. LR is funded by the National Institute of Health (5R01DK124806). We are highly indebted to extend our gratitude to the authors of primary studies for responding to our data enquiries. Finally, librarians from Monash University are also appreciated for their support during accessing freely available studies.

## Author contribution

SL, JJ, LR, KV conceptualised the research question. AF contributed to the search of the articles. SL, MC, JG, NH, LR, JJ, KV, WWT, KL, GGU, SC screened the articles, JG, NH, WWT, GGU, GL, NH, JG, SJZ, RT, MP, KL, MB, EGM, AQ, WH extracted the data and appraised the studies. WWT and SL coded the intervention characteristics and analysed the data. WWT drafted the manuscript. All authors have revised and approved the final version of the manuscript and agreed to be accountable for all aspects of the work.

## Competing interest

The authors declare no competing interests.

## Additional information

[1]Eastern Health Clinical School, Monash University, Melbourne, VIC, Australia. [2]Kaiser Permanente Northwest, Kaiser Permanente Center for Health Research, Oakland, USA. [3]Northwestern University/ Lurie Children's Hospital of Chicago, Chicago, IL, USA. [4]Pennington Biomedical Research Center, Baton Rouge, LA, USA. [5]Madras Diabetes Research Foundation Chennai, Chennai, India. [6]Deakin University, Melbourne, Australia. [7]Department of Nutrition, Dietetics and Food, Monash University, Melbourne, VIC, Australia. [8]Monash Centre for Health Research and Implementation, Monash University, Clayton, VIC, Australia. [9]Department of Women and Children's Health, King's College London, London, UK. [10]Ann & Robert H. Lurie Children's Hospital of Chicago, Chicago, IL, USA. [11]Department of Clinical & Organizational Development, University of Chicago, Chicago, IL, USA. [12]Adelaide Medical School, Faculty of Health and Medical Sciences, The University of Adelaide, Adelaide, Australia. [13]Global Health Institute, University of Antwerp, Antwerp, Belgium. [14]School of Health Sciences, University of Newcastle, Callaghan, NSW, Australia. [15]School of Agriculture, Food and Wine, The University of Adelaide, Adelaide, Australia. ✉e-mail: siew.lim1@monash.edu

## ADA/EASD PMDI

Deirdre K. Tobias[16,17], Jordi Merino[18,19,20], Abrar Ahmad[21], Catherine Aiken[22,23], Jamie L. Benham[24], Dhanasekaran Bodhini[25], Amy L. Clark[26], Kevin Colclough[27], Rosa Corcoy[28,29,30], Sara J. Cromer[19,31,32], Daisy Duan[33], Jamie L. Felton[34,35,36], Ellen C. Francis[37], Pieter Gillard[38], Véronique Gingras[39,40], Romy Gaillard[41], Eram Haider[42], Alice Hughes[27], Jennifer M. Ikle[43,44], Laura M. Jacobsen[45], Anna R. Kahkoska[46], Jarno L. T. Kettunen[47,48,49], Raymond J. Kreienkamp[19,20,31,50], Lee-Ling Lim[51,52,53], Jonna M. E. Männistö[54,55], Robert Massey[42], Niamh-Maire Mclennan[56], Rachel G. Miller[57], Mario Luca Morieri[58,59], Jasper Most[60], Rochelle N. Naylor[61], Bige Ozkan[62,63], Kashyap Amratlal Patel[27], Scott J. Pilla[64,65], Katsiaryna Prystupa[66,67], Sridharan Raghavan[68,69], Mary R. Rooney[62,70], Martin Schön[66,67,71], Zhila Semnani-Azad[17], Magdalena Sevilla-Gonzalez[31,32,72], Pernille Svalastoga[73,74], Wubet Worku Takele[1], Claudia Ha-ting Tam[53,75,76], Anne Cathrine B. Thuesen[18], Mustafa Tosur[77,78,79], Amelia S. Wallace[62,70], Caroline C. Wang[70], Jessie J. Wong[80], Jennifer M. Yamamoto[81], Katherine Young[27], Chloé Amouyal[82,83], Mette K. Andersen[18], Maxine P. Bonham[7], Mingling Chen [8], Feifei Cheng[84], Tinashe Chikowore[32,85,86,87], Sian C. Chivers [9], Christoffer Clemmensen[18], Dana Dabelea[88], Adem Y. Dawed[42], Aaron J. Deutsch[20,31,32], Laura T. Dickens[89], Linda A. DiMeglio[34,35,36,90], Monika Dudenhöffer-Pfeifer[21], Carmella Evans-Molina[34,35,36,91], María Mercè Fernández-Balsells[92,93], Hugo Fitipaldi[21], Stephanie L. Fitzpatrick[94], Stephen E. Gitelman[95], Mark O. Goodarzi[96,97], Jessica A. Grieger[12,98], Marta Guasch-Ferré[17,99], Nahal Habibi[12,98], Torben Hansen[18], Chuiguo Huang[53,75], Arianna Harris-Kawano[34,35,36], Heba M. Ismail[34,35,36], Benjamin Hoag[100,101], Randi K. Johnson[102,103], Angus G. Jones[27,104], Robert W. Koivula[105], Aaron Leong[19,32,106], Gloria K. W. Leung[7], Ingrid M. Libman[107], Kai Liu[12], S. Alice Long[108], William L. Lowe Jr.[109], Robert W. Morton[110,111,112], Ayesha A. Motala[113], Suna Onengut-Gumuscu[114], James S. Pankow[115], Maleesa Pathirana[12,98], Sofia Pazmino[116], Dianna Perez[34,35,36], John R. Petrie[117], Camille E. Powe[19,31,32,118], Alejandra Quinteros [12], Rashmi Jain[119,120], Debashree Ray[70,121], Mathias Ried-Larsen[122,123], Zeb Saeed[124], Vanessa Santhakumar[16], Sarah Kanbour[64,125], Sudipa Sarkar[64], Gabriela S. F. Monaco[34,35,36], Denise M. Scholtens[126], Elizabeth Selvin[62,70], Wayne Huey-Herng Sheu[127,128,129], Cate Speake[130], Maggie A. Stanislawski[102], Nele Steenackers[116], Andrea K. Steck[131], Norbert Stefan[67,132,133], Julie Støy[134], Rachael Taylor[135], Sok Cin Tye[136,137], Gebresilasea Gendisha Ukke[1], Marzhan Urazbayeva[78,138], Bart Van der Schueren[116,139], Camille Vatier[140,141], John M. Wentworth[142,143,144], Wesley Hannah[6,145], Sara L. White[9,146], Gechang Yu[53,75], Yingchai Zhang[53,75], Shao J. Zhou[98,147], Jacques Beltrand[148,149], Michel Polak[148,149], Ingvild Aukrust[73,150], Elisa de Franco[27], Sarah E. Flanagan[27], Kristin A. Maloney[151], Andrew McGovern[27], Janne Molnes[73,150], Mariam Nakabuye[18], Pål Rasmus Njølstad[73,74], Hugo Pomares-Millan[21,152], Michele Provenzano[153], Cécile Saint-Martin[154], Cuilin Zhang[155,156], Yeyi Zhu[157,158], Sungyoung Auh[159], Russell de Souza[111,160], Andrea J. Fawcett[161,162], Chandra Gruber[163], Eskedar Getie Mekonnen[164,165], Emily Mixter[166], Diana Sherifali[111,167], Robert H. Eckel[168], John J. Nolan[169,170], Louis H. Philipson[166], Rebecca J. Brown[159], Liana K. Billings[171,172], Kristen Boyle[88], Tina Costacou[57], John M. Dennis[27], Jose C. Florez[19,20,31,32], Anna L. Gloyn[43,44,173], Maria F. Gomez[21,174],

Peter A. Gottlieb[131], Siri Atma W. Greeley[175], Kurt Griffin[120,176], Andrew T. Hattersley[27,104], Irl B. Hirsch[177], Marie-France Hivert[19,178,179], Korey K. Hood[80], Jami L. Josefson[161], Soo Heon Kwak[180], Lori M. Laffel[181], Siew S. Lim[1], Ruth J. F. Loos[18,182], Ronald C. W. Ma[53,75,76], Chantal Mathieu[38], Nestoras Mathioudakis[64], James B. Meigs[32,106,183], Shivani Misra[184,185], Viswanathan Mohan[186], Rinki Murphy[187,188,189], Richard Oram[27,104], Katharine R. Owen[105,190], Susan E. Ozanne[191], Ewan R. Pearson[42], Wei Perng[88], Toni I. Pollin[151,192], Rodica Pop-Busui[193], Richard E. Pratley[194], Leanne M. Redman[4], Maria J. Redondo[77,78], Rebecca M. Reynolds[56], Robert K. Semple[56,195], Jennifer L. Sherr[196], Emily K. Sims[34,35,36], Arianne Sweeting[197,198], Tiinamaija Tuomi[47,48,49], Miriam S. Udler[19,20,31,32], Kimberly K. Vesco[2], Tina Vilsbøll[199,200], Robert Wagner[66,67,201], Stephen S. Rich[114] & Paul W. Franks[17,21,105,112]

[16]Division of Preventative Medicine, Department of Medicine, Brigham and Women's Hospital and Harvard Medical School, Boston, MA, USA. [17]Department of Nutrition, Harvard T.H. Chan School of Public Health, Boston, MA, USA. [18]Novo Nordisk Foundation Center for Basic Metabolic Research, Faculty of Health and Medical Sciences, University of Copenhagen, Copenhagen, Denmark. [19]Diabetes Unit, Endocrine Division, Massachusetts General Hospital, Boston, MA, USA. [20]Center for Genomic Medicine, Massachusetts General Hospital, Boston, MA, USA. [21]Department of Clinical Sciences, Lund University Diabetes Centre, Lund University, Malmö, Sweden. [22]Department of Obstetrics and Gynaecology, The Rosie Hospital, Cambridge, UK. [23]NIHR Cambridge Biomedical Research Centre, University of Cambridge, Cambridge, UK. [24]Departments of Medicine and Community Health Sciences, Cumming School of Medicine, University of Calgary, Calgary, AB, Canada. [25]Department of Molecular Genetics, Madras Diabetes Research Foundation, Chennai, India. [26]Division of Pediatric Endocrinology, Department of Pediatrics, Saint Louis University School of Medicine, SSM Health Cardinal Glennon Children's Hospital, St. Louis, MO, USA. [27]Department of Clinical and Biomedical Sciences, University of Exeter Medical School, Devon, UK. [28]CIBER-BBN, ISCIII, Madrid, Spain. [29]Institut d'Investigació Biomèdica Sant Pau (IIB SANT PAU), Barcelona, Spain. [30]Departament de Medicina, Universitat Autònoma de Barcelona, Bellaterra, Spain. [31]Programs in Metabolism and Medical & Population Genetics, Broad Institute, Cambridge, MA, USA. [32]Department of Medicine, Harvard Medical School, Boston, MA, USA. [33]Division of Endocrinology, Diabetes and Metabolism, Johns Hopkins University School of Medicine, Baltimore, MD, USA. [34]Department of Pediatrics, Indiana University School of Medicine, Indianapolis, IN, USA. [35]Herman B Wells Center for Pediatric Research, Indiana University School of Medicine, Indianapolis, IN, USA. [36]Center for Diabetes and Metabolic Diseases, Indiana University School of Medicine, Indianapolis, IN, USA. [37]Department of Biostatistics and Epidemiology, Rutgers School of Public Health, Piscataway, NJ, USA. [38]University Hospital Leuven, Leuven, Belgium. [39]Department of Nutrition, Université de Montréal, Montreal, QC, Canada. [40]Research Center, Sainte-Justine University Hospital Center, Montreal, QC, Canada. [41]Department of Pediatrics, Erasmus Medical Center, Rotterdam, The Netherlands. [42]Division of Population Health & Genomics, School of Medicine, University of Dundee, Dundee, UK. [43]Department of Pediatrics, Stanford School of Medicine, Stanford University, CA, USA. [44]Stanford Diabetes Research Center, Stanford School of Medicine, Stanford University, CA, USA. [45]University of Florida, Gainesville, FL, USA. [46]Department of Nutrition, University of North Carolina at Chapel Hill, Chapel Hill, NC, USA. [47]Helsinki University Hospital, Abdominal Centre/Endocrinology, Helsinki, Finland. [48]Folkhalsan Research Center, Helsinki, Finland. [49]Institute for Molecular Medicine Finland FIMM, University of Helsinki, Helsinki, Finland. [50]Department of Pediatrics, Division of Endocrinology, Boston Children's Hospital, Boston, MA, USA. [51]Department of Medicine, Faculty of Medicine, University of Malaysia, Kuala Lumpur, Malaysia. [52]Asia Diabetes Foundation, Hong Kong, SAR, China. [53]Department of Medicine & Therapeutics, Chinese University of Hong Kong, Hong Kong, SAR, China. [54]Departments of Pediatrics and Clinical Genetics, Kuopio University Hospital, Kuopio, Finland. [55]Department of Medicine, University of Eastern Finland, Kuopio, Finland. [56]Centre for Cardiovascular Science, Queen's Medical Research Institute, University of Edinburgh, Edinburgh, UK. [57]Department of Epidemiology, University of Pittsburgh, Pittsburgh, PA, USA. [58]Metabolic Disease Unit, University Hospital of Padova, Padova, Italy. [59]Department of Medicine, University of Padova, Padova, Italy. [60]Department of Orthopedics, Zuyderland Medical Center, Sittard-Geleen, The Netherlands. [61]Departments of Pediatrics and Medicine, University of Chicago, Chicago, IL, USA. [62]Welch Center for Prevention, Epidemiology, and Clinical Research, Johns Hopkins Bloomberg School of Public Health, Baltimore, MD, USA. [63]Ciccarone Center for the Prevention of Cardiovascular Disease, Johns Hopkins School of Medicine, Baltimore, MD, USA. [64]Department of Medicine, Johns Hopkins University, Baltimore, MD, USA. [65]Department of Health Policy and Management, Johns Hopkins University Bloomberg School of Public Health, Baltimore, MD, USA. [66]Institute for Clinical Diabetology, German Diabetes Center, Leibniz Center for Diabetes Research at Heinrich Heine University Düsseldorf, Auf'm Hennekamp 65, 40225 Düsseldorf, Germany. [67]German Center for Diabetes Research (DZD), Ingolstädter Landstraße 1, 85764 Neuherberg, Germany. [68]Section of Academic Primary Care, US Department of Veterans Affairs Eastern Colorado Health Care System, Aurora, CO, USA. [69]Department of Medicine, University of Colorado School of Medicine, Aurora, CO, USA. [70]Department of Epidemiology, Johns Hopkins Bloomberg School of Public Health, Baltimore, MD, USA. [71]Institute of Experimental Endocrinology, Biomedical Research Center, Slovak Academy of Sciences, Bratislava, Slovakia. [72]Clinical and Translational Epidemiology Unit, Massachusetts General Hospital, Boston, MA, USA. [73]Mohn Center for Diabetes Precision Medicine, Department of Clinical Science, University of Bergen, Bergen, Norway. [74]Children and Youth Clinic, Haukeland University Hospital, Bergen, Norway. [75]Laboratory for Molecular Epidemiology in Diabetes, Li Ka Shing Institute of Health Sciences, The Chinese University of Hong Kong, Hong Kong, China. [76]Hong Kong Institute of Diabetes and Obesity, The Chinese University of Hong Kong, Hong Kong, China. [77]Department of Pediatrics, Baylor College of Medicine, Houston, TX, USA. [78]Division of Pediatric Diabetes and Endocrinology, Texas Children's Hospital, Houston, TX, USA. [79]Children's Nutrition Research Center, USDA/ARS, Houston, TX, USA. [80]Stanford University School of Medicine, Stanford, CA, USA. [81]Internal Medicine, University of Manitoba, Winnipeg, MB, Canada. [82]Department of Diabetology, APHP, Paris, France. [83]Sorbonne Université, INSERM, NutriOmic team, Paris, France. [84]Health Management Center, The Second Affiliated Hospital of Chongqing Medical University, Chongqing Medical University, Chongqing, China. [85]MRC/Wits Developmental Pathways for Health Research Unit, Department of Paediatrics, Faculty of Health Sciences, University of the Witwatersrand, Johannesburg, South Africa. [86]Channing Division of Network Medicine, Brigham and Women's Hospital, Boston, MA, USA. [87]Sydney Brenner Institute for Molecular Bioscience, Faculty of Health Sciences, University of the Witwatersrand, Johannesburg, South Africa. [88]Lifecourse Epidemiology of Adiposity and Diabetes (LEAD) Center, University of Colorado Anschutz Medical Campus, Aurora, CO, USA. [89]Section of Adult and Pediatric Endocrinology, Diabetes and Metabolism, Kovler Diabetes Center, University of Chicago, Chicago, USA. [90]Department of Pediatrics, Riley Hospital for Children, Indiana University School of Medicine, Indianapolis, IN, USA. [91]Richard L. Roudebush VAMC, Indianapolis, IN, USA. [92]Biomedical Research Institute Girona, IdIBGi, Girona, Spain. [93]Diabetes, Endocrinology and Nutrition Unit Girona, University Hospital Dr Josep Trueta, Girona, Spain. [94]Institute of Health System Science, Feinstein Institutes for Medical Research, Northwell Health, Manhasset, NY, USA. [95]University of California at San Francisco, Department of Pediatrics, Diabetes Center, San Francisco, CA, USA. [96]Division of Endocrinology, Diabetes and Metabolism, Cedars-Sinai Medical Center, Los Angeles, CA, USA. [97]Department of Medicine, Cedars-Sinai Medical Center, Los Angeles, CA, USA. [98]Robinson Research Institute, The University of Adelaide, Adelaide, Australia. [99]Department of Public Health and Novo Nordisk Foundation Center for Basic Metabolic Research, Faculty of Health and Medical Sciences, University of Copenhagen, 1014 Copenhagen, Denmark. [100]Division of Endocrinology and Diabetes, Department of Pediatrics, Sanford Children's Hospital, Sioux Falls, SD, USA. [101]University of South Dakota School of Medicine, E Clark St, Vermillion, SD, USA. [102]Department of Biomedical Informatics, University of Colorado Anschutz Medical Campus, Aurora, CO, USA. [103]Department of Epidemiology, Colorado School of Public Health, Aurora, CO, USA. [104]Royal Devon University Healthcare NHS Foundation Trust, Exeter, UK. [105]Oxford Centre for Diabetes, Endocrinology and Metabolism, University of Oxford, Oxford, UK. [106]Division of General Internal Medicine, Massachusetts General Hospital, Boston,

MA, USA. [107]UPMC Children's Hospital of Pittsburgh, Pittsburgh, PA, USA. [108]Center for Translational Immunology, Benaroya Research Institute, Seattle, WA, USA. [109]Department of Medicine, Northwestern University Feinberg School of Medicine, Chicago, IL, USA. [110]Department of Pathology & Molecular Medicine, McMaster University, Hamilton, Canada. [111]Population Health Research Institute, Hamilton, Canada. [112]Department of Translational Medicine, Medical Science, Novo Nordisk Foundation, Tuborg Havnevej 19, 2900 Hellerup, Denmark. [113]Department of Diabetes and Endocrinology, Nelson R Mandela School of Medicine, University of KwaZulu-Natal, Durban, South Africa. [114]Center for Public Health Genomics, Department of Public Health Sciences, University of Virginia, Charlottesville, VA, USA. [115]Division of Epidemiology and Community Health, School of Public Health, University of Minnesota, Minneapolis, MN, USA. [116]Department of Chronic Diseases and Metabolism, Clinical and Experimental Endocrinology, KU Leuven, Leuven, Belgium. [117]School of Health and Wellbeing, College of Medical, Veterinary and Life Sciences, University of Glasgow, Glasgow, UK. [118]Department of Obstetrics, Gynecology, and Reproductive Biology, Massachusetts General Hospital and Harvard Medical School, Boston, MA, USA. [119]Sanford Children's Specialty Clinic, Sioux Falls, SD, USA. [120]Department of Pediatrics, Sanford School of Medicine, University of South Dakota, Sioux Falls, SD, USA. [121]Department of Biostatistics, Johns Hopkins Bloomberg School of Public Health, Baltimore, MD, USA. [122]Centre for Physical Activity Research, Rigshospitalet, Copenhagen, Denmark. [123]Institute for Sports and Clinical Biomechanics, University of Southern Denmark, Odense, Denmark. [124]Department of Medicine, Division of Endocrinology, Diabetes and Metabolism, Indiana University School of Medicine, Indianapolis, IN, USA. [125]AMAN Hospital, Doha, Qatar. [126]Department of Preventive Medicine, Division of Biostatistics, Northwestern University Feinberg School of Medicine, Chicago, IL, USA. [127]Institute of Molecular and Genomic Medicine, National Health Research Institutes, Taipei, Taiwan. [128]Division of Endocrinology and Metabolism, Taichung Veterans General Hospital, Taichung, Taiwan. [129]Division of Endocrinology and Metabolism, Taipei Veterans General Hospital, Taipei, Taiwan. [130]Center for Interventional Immunology, Benaroya Research Institute, Seattle, WA, USA. [131]Barbara Davis Center for Diabetes, University of Colorado Anschutz Medical Campus, Aurora, CO, USA. [132]University Hospital of Tübingen, Tübingen, Germany. [133]Institute of Diabetes Research and Metabolic Diseases (IDM), Helmholtz Center Munich, Neuherberg, Germany. [134]Steno Diabetes Center Aarhus, Aarhus University Hospital, Aarhus, Denmark. [135]University of Newcastle, Newcastle upon Tyne, UK. [136]Sections on Genetics and Epidemiology, Joslin Diabetes Center, Harvard Medical School, Boston, MA, USA. [137]Department of Clinical Pharmacy and Pharmacology, University Medical Center Groningen, Groningen, The Netherlands. [138]Gastroenterology, Baylor College of Medicine, Houston, TX, USA. [139]Department of Endocrinology, University Hospitals Leuven, Leuven, Belgium. [140]Sorbonne University, Inserm U938, Saint-Antoine Research Centre, Institute of Cardiometabolism and Nutrition, Paris 75012, France. [141]Department of Endocrinology, Diabetology and Reproductive Endocrinology, Assistance Publique-Hôpitaux de Paris, Saint-Antoine University Hospital, National Reference Center for Rare Diseases of Insulin Secretion and Insulin Sensitivity (PRISIS), Paris, France. [142]Royal Melbourne Hospital Department of Diabetes and Endocrinology, Parkville, VIC, Australia. [143]Walter and Eliza Hall Institute, Parkville, VIC, Australia. [144]University of Melbourne Department of Medicine, Parkville, VIC, Australia. [145]Department of Epidemiology, Madras Diabetes Research Foundation, Chennai, India. [146]Department of Diabetes and Endocrinology, Guy's and St Thomas' Hospitals NHS Foundation Trust, London, UK. [147]School of Agriculture, Food and Wine, University of Adelaide, Adelaide, Australia. [148]Institut Cochin, Inserm U, 10116 Paris, France. [149]Pediatric endocrinology and diabetes, Hopital Necker Enfants Malades, APHP Centre, université de Paris, Paris, France. [150]Department of Medical Genetics, Haukeland University Hospital, Bergen, Norway. [151]Department of Medicine, University of Maryland School of Medicine, Baltimore, MD, USA. [152]Department of Epidemiology, Geisel School of Medicine at Dartmouth, Hanover, NH, USA. [153]Nephrology, Dialysis and Renal Transplant Unit, IRCCS—Azienda Ospedaliero-Universitaria di Bologna, Alma Mater Studiorum University of Bologna, Bologna, Italy. [154]Department of Medical Genetics, AP-HP Pitié-Salpêtrière Hospital, Sorbonne University, Paris, France. [155]Global Center for Asian Women's Health, Yong Loo Lin School of Medicine, National University of Singapore, Singapore, Singapore. [156]Department of Obstetrics and Gynecology, Yong Loo Lin School of Medicine, National University of Singapore, Singapore, Singapore. [157]Kaiser Permanente Northern California Division of Research, Oakland, CA, USA. [158]Department of Epidemiology and Biostatistics, University of California San Francisco, San Francisco, CA, USA. [159]National Institute of Diabetes and Digestive and Kidney Diseases, National Institutes of Health, Bethesda, MD, USA. [160]Department of Health Research Methods, Evidence, and Impact, Faculty of Health Sciences, McMaster University, Hamilton, ON, Canada. [161]Ann & Robert H. Lurie Children's Hospital of Chicago, Department of Pediatrics, Northwestern University Feinberg School of Medicine, Chicago, IL, USA. [162]Department of Clinical and Organizational Development, Chicago, IL, USA. [163]American Diabetes Association, Arlington, VI, USA. [164]College of Medicine and Health Sciences, University of Gondar, Gondar, Ethiopia. [165]Global Health Institute, Faculty of Medicine and Health Sciences, University of Antwerp, 2160 Antwerp, Belgium. [166]Department of Medicine and Kovler Diabetes Center, University of Chicago, Chicago, IL, USA. [167]School of Nursing, Faculty of Health Sciences, McMaster University, Hamilton, Canada. [168]Division of Endocrinology, Metabolism, Diabetes, University of Colorado, Colorado, CO, USA. [169]Department of Clinical Medicine, School of Medicine, Trinity College Dublin, Dublin, Ireland. [170]Department of Endocrinology, Wexford General Hospital, Wexford, Ireland. [171]Division of Endocrinology, NorthShore University HealthSystem, Skokie, IL, USA. [172]Department of Medicine, Prtizker School of Medicine, University of Chicago, Chicago, IL, USA. [173]Department of Genetics, Stanford School of Medicine, Stanford University, CA, USA. [174]Faculty of Health, Aarhus University, Aarhus, Denmark. [175]Departments of Pediatrics and Medicine and Kovler Diabetes Center, University of Chicago, Chicago, USA. [176]Sanford Research, Sioux Falls, SD, USA. [177]University of Washington School of Medicine, Seattle, WA, USA. [178]Department of Population Medicine, Harvard Medical School, Harvard Pilgrim Health Care Institute, Boston, MA, USA. [179]Department of Medicine, Universite de Sherbrooke, Sherbrooke, QC, Canada. [180]Department of Internal Medicine, Seoul National University College of Medicine, Seoul National University Hospital, Seoul, Republic of Korea. [181]Joslin Diabetes Center, Harvard Medical School, Boston, MA, USA. [182]Charles Bronfman Institute for Personalized Medicine, Icahn School of Medicine at Mount Sinai, New York, NY, USA. [183]Broad Institute, Cambridge, MA, USA. [184]Division of Metabolism, Digestion and Reproduction, Imperial College London, London, UK. [185]Department of Diabetes & Endocrinology, Imperial College Healthcare NHS Trust, London, UK. [186]Department of Diabetology, Madras Diabetes Research Foundation & Dr. Mohan's Diabetes Specialities Centre, Chennai, India. [187]Department of Medicine, Faculty of Medicine and Health Sciences, University of Auckland, Auckland, New Zealand. [188]Auckland Diabetes Centre, Te Whatu Ora Health New Zealand, Auckland, New Zealand. [189]Medical Bariatric Service, Te Whatu Ora Counties, Health New Zealand, Auckland, New Zealand. [190]Oxford NIHR Biomedical Research Centre, University of Oxford, Oxford, UK. [191]University of Cambridge, Metabolic Research Laboratories and MRC Metabolic Diseases Unit, Wellcome-MRC Institute of Metabolic Science, Cambridge, UK. [192]Department of Epidemiology & Public Health, University of Maryland School of Medicine, Baltimore, MD, USA. [193]Department of Internal Medicine, Division of Metabolism, Endocrinology and Diabetes, University of Michigan, Ann Arbor, MI, USA. [194]AdventHealth Translational Research Institute, Orlando, FL, USA. [195]MRC Human Genetics Unit, Institute of Genetics and Cancer, University of Edinburgh, Edinburgh, UK. [196]Yale School of Medicine, New Haven, CT, USA. [197]Faculty of Medicine and Health, University of Sydney, Sydney, NSW, Australia. [198]Department of Endocrinology, Royal Prince Alfred Hospital, Sydney, NSW, Australia. [199]Clinial Research, Steno Diabetes Center Copenhagen, Herlev, Denmark. [200]Department of Clinical Medicine, Faculty of Health and Medical Sciences, University of Copenhagen, Copenhagen, Denmark. [201]Department of Endocrinology and Diabetology, University Hospital Düsseldorf, Heinrich Heine University Düsseldorf, Moorenstr. 5, 40225 Düsseldorf, Germany.

