## [Peer Review File · Communications Medicine]

Reviewers' comments:

Reviewer #1 (Remarks to the Author):

This is a really comprehensive and novel systematic review and meta analysis evaluating the effect of different interventions for GDM prevention during pregnancy. The findings from this work is of high interest to pregnant people worldwide given GDM is one of the most common complications during pregnancy. In general, the manuscript is written very succinctly and easy to follow. The systematic review and meta-analysis is well justified in the introduction section.

Major comments:

Methods section -

1. Line 138, it would be helpful to have detailed date on when was the last search performed.
2. Line 152-154, how was the screening process conducted? Like have you used any tools or platform to manage the large number of records? Which authors participated in title/abstract vs. full-text review stage?

Results section -

3. Line 267-277, do you have any data on what kind of physical activity interventions were included, are they walking only, aerobic or resistance training. Right now you treated all physical activity intervention as the same which might be a limitation of your review.. Also, are the physical activity interventions delivered at healthcare facilities having better attendance or adherence? These might be confounding factors for the delivery format.

Discussion section -

4. Line 357-359, given what you have identified in the introduction section as a gap (Line 95), it would be helpful to discuss these main findings in the discussion section. It is meaningful that most of the studied intervention components (e.g., physical activity, diet, metformin, ...) are effective for GDM prevention, and it has significant implications to the community.
5. Line 367-418, for the discussions on physical activity, again, I would like to see some discussion on different types, intensity and/or duration of physical activity. If it is outside the scope of current review, at least point out for future research to look at these directions for physical activities during pregnancy.

Conclusion

6. Before going to the details about the characteristics of more effective physical activity intervention, it is equally important to mention the effective intervention components identified by this comprehensive systematic review and meta analysis

Minor comments:

1. abstract line 41 - "This evaluated intervention components...", do you mean "This REVIEW evaluated intervention components..."
2. line 232, remove the first coma ",," before "investing lifestyle..."
3. Line 456, consider adding "during pregnancy" before "... significantly reduce..."
4. For the reference list, the format is inconsistent, e.g., the title of the journals
5. Supplement table 4, it is not clear what do the numbers in parenthesis in column 1 (after study and year) for. Also, there was a line with only a "Yes" in the last column and the rest of the row is

blank.

Reviewer #2 (Remarks to the Author):

This is a rigorous systematic review and meta-analysis that examines the effect of various interventions on risk of incident gestational diabetes. This review is registered with Prospero, and follows appropriate guidelines including PRISMA, GRADE, and TIDieR. The authors have synthesized the vast amount of information well. I have some comments for consideration, detailed below.

1. Did the authors consider including a search of clinical trial databases? This may mitigate the publication bias reported throughout.
2. Conclusions re low-income settings are based on a small number of studies and so I would temper conclusions about this (i.e. perhaps this is not a key conclusion to summarize at the end of your article)
3. Forest plots should be in main article for key results, not in supplement

Reviewer #1 (Remarks to the Author):

This is a really comprehensive and novel systematic review and meta-analysis evaluating the effect of different interventions for GDM prevention during pregnancy. The findings from this work is of high interest to pregnant people worldwide given GDM is one of the most common complications during pregnancy. In general, the manuscript is written very succinctly and easy to follow. The systematic review and meta-analysis is well justified in the introduction section.

Thank you for your comments. Responses to each concern are provided below.

Methods section -

1. Line 138, it would be helpful to have detailed date on when was the last search performed.

Response: As documented in line 108, the last search was done on May 24, 2022.

2. Line 152-154, how was the screening process conducted? Like have you used any tools or platform to manage the large number of records? Which authors participated in title/abstract vs. full-text review stage?

Response: Acknowledging your insight, the description featured below has been added to the main document from line 115-119.

The retrieved articles from several databases were exported to Endnote version 20 (Clarivate), and duplicates were removed. Hand searches, including the reference list of related reviews, were also assessed for additional eligible studies. Covidence (Veritas Health Innovation, Melbourne, Australia), an online software was used for title/abstract screening and full-text reviews.

Two reviewers from the reviewers' team (WWT, JG, NH, GGU, GL, SJZ, RT, MP, KL, MB, and AQ) independently screened each record for eligibility, and disagreements were resolved by discussion with an arbiter (SL).

Results section -

3. Line 267-277, do you have any data on what kind of physical activity interventions were included, are they walking only, aerobic or resistance training. Right now you treated all physical activity intervention as the same which might be a limitation of your review. Also,

are the physical activity interventions delivered at healthcare facilities having better attendance or adherence? These might be confounding factors for the delivery format.

Response: Following your suggestion, we have done additional subgroup analysis by the type of physical activity (walking-only Vs aerobic Vs aerobic and resistance) and duration of the session (<45 minutes Vs ≥45 minutes). There was no difference in the type and duration of physical activity in the effectiveness of preventing GDM. See Table 2 for the subgroup findings and Supplementary Table 9 for the description of each study.

We found that physical activity interventions delivered in healthcare facilities prevent GDM more effectively than in other settings. One of the possible reasons for better effectiveness of interventions delivered in healthcare facilities could be associated with better supervision and adherence of participants to the intervention. However, data on adherence were usually not reported. We have commented on this limitation in the discussion section from lines 350-355:

However, as data on the level of adherence to physical activity interventions delivered in different settings were not reported, it is impossible to draw inference that the better effectiveness of healthcare facility-based interventions are related to the better adherence of participants to the intervention. Future primary studies are recommended to examine the role of adherence rate in the effectiveness of physical activity interventions delivered in different settings.

Discussion section -

4. Line 357-359, given what you have identified in the introduction section as a gap (Line 95), it would be helpful to discuss these main findings in the discussion section. It is meaningful that most of the studied intervention components (e.g., physical activity, diet, metformin, ...) are effective for GDM prevention, and it has significant implications to the community.

Response: We have added some statements as described below. See line from 302-309.

The findings are in line with the most recent findings from umbrella reviews, implying the significant importance of incorporating these interventions in routine maternal care to prevent GDM. However, the primary analysis of this review has previously shown that not all interventions work equally for all participants, and therefore, considering person-level characteristics (e.g., previous history of GDM) during implementation could be important to

enhance the effectiveness of interventions. This secondary analysis shows the differences in the intervention effectiveness by intervention type and delivery.

5. Line 367-418, for the discussions on physical activity, again, I would like to see some discussion on different types, intensity and/or duration of physical activity. If it is outside the scope of current review, at least point out for future research to look at these directions for physical activities during pregnancy.

Responses: Discussion on physical activity type is as stated in our previous comment above. We have now conducted subgroup analysis by the level of intensity of physical activity, and we did not find significant differences by intensity. We have added the statements featured below in the discussion section. See from line 374-383.

Moreover, significant differences in the effectiveness of physical activity intervention across other intervention characteristics, including intensity, session duration, and type of physical activity, were not observed. Similar to a recent umbrella review (103), we found that physical activity interventions of light-moderate or moderate intensity effectively reduced the risk of GDM. However, the differences between subgroups by intensity were found to be insignificant (p-value=0.18). It was evidenced that light to moderate or moderate intensity reduced the incidence of GDM compared with moderate to vigorous intensity (103). Given the effectiveness of light-moderate activities, which are more achievable than higher-intensity training, especially during pregnancy, women at risk of GDM should be recommended to engage in moderate intensity activities to reduce their GDM risk.

Considering your suggestion, we have also undertaken subgroup analysis by session duration, and we didn't find a significant difference.

Conclusions

6. Before going to the details about the characteristics of more effective physical activity intervention, it is equally important to mention the effective intervention components identified by this comprehensive systematic review and meta-analysis.

Response: We have added the statement featured below. See line 321-323.

Dietary, physical activity, diet plus physical activity, metformin, and myoinositol interventions during pregnancy significantly reduced the incidence of GDM compared with control interventions.

We also noticed a potential confusion for readers between the term ‘component’ and ‘characteristic’. Component refers to the various interventions in a multi-modal intervention (eg metformin plus diet), whilst characteristic refers to the descriptive nature of the intervention as specified in TIDieR (eg online or in-person). The primary aim of the current review is on the latter, as the difference between the intervention component or types were previously covered in the primary analysis of this review (105). We have changed ‘components’ to ‘characteristics’ in this review for clarity.

Minor comments:

1. abstract line 41 - "This evaluated intervention components...", do you mean "This REVIEW evaluated intervention components..."

Response: We have added the term ‘review’(line 34). Current version: ‘This review evaluated intervention characteristics to inform the implementation of GDM prevention interventions’.

2. line 232, remove the first coma ",," before "investing lifestyle..."

Response: A comma has been inserted (line 200). This now reads, ‘of the 92 included studies investigating lifestyle intervention, 59(64.1%) included combined physical activity and dietary interventions...’

3. Line 456, consider adding "during pregnancy" before "... significantly reduce..."

Response: The suggestion has been added (line 423-424). This now read as, ‘Group and healthcare facility-based physical activity interventions during pregnancy significantly reduce....’

4. For the reference list, the format is inconsistent, e.g., the title of the journals

Response: edition has been done to ensure the proper formatting.

5. Supplement table 4, it is not clear what do the numbers in parenthesis in column 1 (after study and year) for. Also, there was a line with only a "Yes" in the last column and the rest of the row is blank.

Response: The numbers in brackets were citations, but we have now removed all the numbers. The empty row was mistakenly inserted, and we have deleted that.

Reviewer #2 (Remarks to the Author):

This is a rigorous systematic review and meta-analysis that examines the effect of various interventions on risk of incident gestational diabetes. This review is registered with Prospero, and follows appropriate guidelines including PRISMA, GRADE, and TIDieR. The authors have synthesized the vast amount of information well. I have some comments for consideration, detailed below.

1. Did the authors consider including a search of clinical trial databases? This may mitigate the publication bias reported throughout.

Response: We did not search clinical trial databases in this large review, as the database is primarily created to register trial protocols, not full-length studies. Records in clinical trial registers do not contain sufficient results for analysis and appraisal. This, therefore, does not mitigate the problem of publication bias.

2. Conclusions re low-income settings are based on a small number of studies and so I would temper conclusions about this (i.e. perhaps this is not a key conclusion to summarize at the end of your article).

Response: Agreed, and we have deleted that.

3. Forest plots should be in main article for key results, not in supplement.

Response: The comment has been accepted, and we have embedded the forest plots in the main document.

REVIEWERS' COMMENTS:

Reviewer #1 (Remarks to the Author):

Thanks for your effort revising the manuscript. Most of my concerns are well addressed. Since you clarified about the two words "components" and "characteristics", I think it is more appropriate to replace "components" with "characteristics" in your title which is the focus of the review.

Reviewer #2 (Remarks to the Author):

thanks for addressing previous comments

Editor's comments

Appreciating your effort to make our study more readable, we have made changes based on your comments and suggestions.

Reviewer #1

Thanks for your effort revising the manuscript. Most of my concerns are well addressed. Since you clarified about the two words "components" and "characteristics", I think it is more appropriate to replace "components" with "characteristics" in your title which is the focus of the review.

Response: Comment accepted and revised accordingly.

Reviewer #2

No comments were given.